# Promising Treatment for Multiple Sclerosis: Mitochondrial Transplantation

**DOI:** 10.3390/ijms23042245

**Published:** 2022-02-17

**Authors:** Pasquale Picone, Domenico Nuzzo

**Affiliations:** Istituto per la Ricerca e l’Innovazione Biomedica, CNR, Via U. La Malfa 153, 90146 Palermo, Italy

**Keywords:** mitochondria, multiple sclerosis, mitochondrial transplantation, biotechnology

## Abstract

In recent years, several studies have examined the multifaceted role of mitochondria in Multiple Sclerosis (MS), suggesting that, besides inflammation and demyelination, mitochondrial aberration is a crucial factor in mediating axonal degeneration, the latter being responsible for persistent disabilities in MS patients. Therefore, mitochondria have been recognized as a possible multiple sclerosis therapeutic target. Recently, mitochondrial transplantation has become a new term for the transfer of live mitochondria into damaged cells for the treatment of various diseases, including neurodegenerative diseases. In this hypothesis, we propose mitochondrial transplantation as a new, potentially applicable approach to counteract axonal degeneration in multiple sclerosis.

## 1. Introduction

Mitochondria are organelles responsible for cellular bioenergetics and play an important role in cellular functions such as calcium homeostasis, reactive-oxygen-species production, cell survival, proliferation, apoptosis, and autophagy. Neurons have many mitochondria, as these cells are highly dependent on oxidative energy metabolism, and mitochondrial dysfunction is known to be involved in several neurodegenerative diseases [1,2]. A knowledge network search for “mitochondria multiple sclerosis” [3], showed that the number of articles has increased significantly over the past two years (2019–2020), indicating that mitochondrial dysfunction has been recognized as a possible important target of MS pathology. In MS patients, multiple studies have provided evidence of mitochondrial dysfunction [4,5], which correlates with axonal degeneration and disease progression [6,7]. Ultrastructural analysis of demyelinated spinal cord lesions showed dramatically reduced numbers of mitochondria, microtubules and axonal swelling. Several independent investigations have demonstrated that molecular changes converge on mitochondria within neurons in MS. The most reproducible changes are related to mitochondrial respiratory chain deficiency, abnormalities in mitochondrial transport and gene expression, oxidative damage, and progressive accumulation of mutations [7] (Figure 1).

The more relevant mechanism contributing to the degeneration of demyelinated axons is an imbalance between the increased energy demand for nerve conduction and the generation of ATP. In neurons, Na^+^/K^+^-ATPase, present in Ranvier’s nodes, creates impulse transmission. In the demyelinated axon, the action of Na^+^/K^+^-ATPase is increased in order to maintain impulse conduction with increased consumption of ATP. When ATP is not readily available, excessive sodium concentration in the axon causes the Na^+^/Ca2^+^ exchanger to operate in reverse, with consequent calcium overload, protease activation and consequent degeneration of the demyelinated axon [8]. Neurons with healthy mitochondria respond to demyelination by increasing the number of mitochondria in acutely demyelinated axons [9], as evidenced in MS autopsy cases and experimental disease models. The increase in mitochondria numbers and activity represents an attempt to alleviate the energy imbalance in the demyelinated axon (Figure 2). Recently, Licht-Mayer and collaborators [10] showed that upon demyelination, mitochondria move from the neuronal cell body to the demyelinated axon, increasing axonal mitochondrial content, a process called the Axonal Response of Mitochondria to Demyelination (ARMD). Interestingly, the enhancement of ARMD, by targeting mitochondrial biogenesis and mitochondrial transport from the cell body to the axon, was shown to protect demyelinated axons from degeneration [10]. Rosenkranz and collaborators showed that boosting the activity of the PGC-1alpha gene, which encodes for PGC-1a (a transcriptional coactivator that acts as a master switch for mitochondrial function), by introducing extra copies of it into neurons, increases numbers of mitochondria and mitochondrial activity (complex IV activity, and maximum respiratory capacity); this makes animals more resistant to the effects of MS [11]. Thus, changes in mitochondrial content and activity in neurons may offer a novel tool to improve neuronal function in patients with MS. Mitochondrial transplantation (MT) is an alternative promising paradigm that may target mitochondrial dysfunction in injury and disease states.

## 2. Mitochondrial Transplantation

Recently, the possibility to transfer healthy mitochondria from one cell to another has represented an attractive therapeutic strategy. With the name of mitochondrial transplantation, it is now indicated that the transfer of live mitochondria into injured cells can treat different diseases, including neurodegenerative diseases [12,13,14,15,16,17]. McCully and colleagues employed mitochondrial transplantation as a therapeutic approach to treat cardiac ischemia in animal models and pediatric patients. Five child patients, aged between 2 days and 2 years old, with cardiac ischemia participated in this study. The mitochondria were extracted during 20–30 min from pieces of samples obtained from the rectus abdominis muscle. The mitochondria were injected using a tuberculin syringe, directly into ischemic areas. In four of the patients, cardiac functions improved, and they were separated from extracorporeal membrane oxygenation support [17]. Although mitochondrial transfer has mainly been utilized for cardiac injuries, this approach has been also applied for treatment of neurodegenerative disease and other injuries of the CNS in animal models [18,19]. Several routes have been used in vivo for MT into the brain, including in situ and systemic approaches. Labelled mitochondria were injected by syringe and stereotactic surgery into the spinal cord of injured rats; the transplanted mitochondria were found within microglia 24 h after the injection [20]. Transplanted mitochondria labelled with an allogenic peptide into the 6-hydroxydopamine (6-OHDA) rat model of Parkinson’s disease (PD), were injected specifically into the medial forebrain bundle. MT induced protective effects on neurons in the nigrostriatal circuit. After a three-month follow up, the motor function of PD rats improved, the mitochondrial function increased, and the cytotoxic effects of 6-OHDA decreased [21]. In another study, mitochondria isolated from hamster cells were injected into the ischemic brain of rats and were able to restore motor function [22]. Apoptotic cells and infarct size significantly decreased in the brain tissue of rats that received mitochondria, revealing that MT has protective effects on neurons after ischemia [22]. Mitochondria delivery to the carotid system through the sonography-guided catheter and intravenous injection are alternative ways to deliver mitochondria to the brain [12]. Two hours after intravenous injection, mitochondria are found in multiple organs, including the brain, resulting in increased ATP content and improved locomotor activity in MPTP-induced Parkinson’s disease mice. Mitochondria can be delivered intravenously due to their small size (~1 µm in diameter), are not incorporated into red blood cells, and do not interfere with the transport of oxygen [23]. Recently, it has been demonstrated that mitochondrial release from the nose to the brain is a feasible approach and is safer than brain injection [24].

Cellular mitochondria internalization was clearly confirmed after 1 h of incubation, and significantly increased after 4 and 24 h. Exogenous mitochondria interact directly with cells, and mechanisms of macropinocytosis (actin-dependent endocytosis) have been involved in mitochondrial cell uptake [25]. Key parameters for the success of MT depend on the source and quality of the isolated mitochondria, mitochondrial stability, an appropriate delivery protocol, and cellular uptake. In clinical practice, the source of mitochondria is the first important step. Skeletal muscles have been suggested as an appropriate source of mitochondria. Specifically, the pectoralis major (in men), rectus abdominis, gastrocnemius, and even neck-strap muscles are suitable sources of mitochondria (Figure 3A). Another source of mitochondria could be spermatozoa [12] (Figure 3A). During isolation, it is essential to meet the criteria of good manufacturing practices (GMP). The size, number, purity, shape, viability, and function of the organelles must be subjected to quality control.

Mitochondria extracted from the patient could be administered in the CNS by different routes (Figure 3B). Intracerebral injection is appropriate for therapeutic interventions, but it is an invasive approach. Intrathecal injection, the delivery of materials to the intrathecal space surrounding the spinal cord, is an alternative fluid-phase delivery route to the CNS. Intrathecal administration causes minimal pain to patients and affords a larger volume of therapeutic materials than intracranial injection. Peripheral administration, systemic or intra-carotid, is a less invasive and safer route, offering the advantages of larger injection volumes and multiple dosages (Figure 3B). The principal obstacles for the intravenous injection are represented by the presence of the blood–brain barrier (BBB), which prevents most drugs from entering the brain, and the diffusion of the drug in total body. Finally, intranasal administration is an alternative route for brain delivery, which bypasses the BBB; however, factors such as limited dosing volume, small absorption surface, the presence of degrading enzymes, and other variables attributed to patient congestion and mucus limit the efficiently. Therefore, to improve the mitochondrial transfer to the brain by a systemic route, biotechnology systems are needed to overcome the BBB or improve intranasal absorption. For example, natural or artificial nanovesicles have been explored as brain-drug delivery systems [26,27].

## 3. Mitochondrial Transfer Technology

MT has been reported as a “magical” cure [26], since healthy mitochondria, harvested by healthy tissue, move to the injured cells after injection, and rescue energy production (ATP) and mitochondrial function. However, MT still presents some weaknesses both in vivo and in vitro. Mitochondria must survive the transition from an intracellular to an inhospitable extracellular environment, and cross cell and body barriers [12,28]. Only a small percentage (10%) of injected mitochondria reaches the cells [17] and the transfer is often not specific to target cells. Therefore, biotechnological approaches are needed to overcome the problems and improve mitochondrial transfer. To improve mitochondrial transfer, conjugation with the carrier peptide Pep-1 (a cell-penetrating peptide that has been employed to facilitate the cellular uptake of nanoparticles, DNA, and proteins) was recently shown to be a valuable method [29]. In addition, a simple and quick protocol was recently introduced for delivering mitochondria to cultured cells, only requiring mitochondria centrifugation at 1500× *g* for 5 min, without additional incubation steps [30].

New strategies have been also recently explored stabilizing mitochondria. The functionalization of the mitochondria surface with hydrophilic biocompatible polymers positively affected transplantation efficiency in both in vitro than in vivo experimentations by stabilizing the colloidal dispersion and reducing phagocytosis by mononuclear cells. In particular, the polysaccharide dextran has been proposed to enhance stabilization without affecting mitochondria activity [31]. In this context, recently, structures obtained from synaptic terminations by physical processes (synaptosomes) have been proposed as vesicles for mitochondrial transfer into neuronal cells [32]. Synaptosomes transport viable mitochondria mainly in the cytoplasm of neuronal cells and restore mitochondrial function in cells containing rotenone-damaged mitochondria [32].

## 4. The Hypothesis of Mitochondrial Transplantation and Multiple Sclerosis

Today, interest in the mitochondrion as a potential target in multiple sclerosis is increasing significantly. Specifically, it is highlighted that an increase in the number and/or activity of mitochondria in the demyelinated neuron can play a neuroprotective role against axonal degeneration. In particular, this hypothesis wants to propose a new, potentially applicable way to counteract axonal degeneration in multiple sclerosis constituted by mitochondrial transplantation. Furthermore, given the critical issues raised, to date, on mitochondrial transplantation, we propose the use and implementation of innovative delivery systems that can allow the transport of mitochondria—protecting them from the inhospitable extracellular environment—into neuronal target cells (Figure 4). In support of this hypothesis Peruzzotti-Jamett and collaborators recently showed first evidence that neural stem cells deliver functional mitochondria to target cells via extracellular vesicles, restoring mitochondrial dysfunction in mice with experimental autoimmune encephalomyelitis, a model of MS [33].

## 5. Conclusions

To date, there is certainly important evidence on the involvement of mitochondria in axonal degeneration in MS, and the evidence that an increase in the number of viable mitochondria, or their activity, could have beneficial effects is beginning to gain strength. An approach that would involve the use of mitochondrial transplantation, to slow down or stop axonal degeneration, is beginning to appear in the scientific landscape, and in vitro and in vivo studies are needed to reinforce this hypothesis.

## Figures and Tables

**Figure 1 ijms-23-02245-f001:**
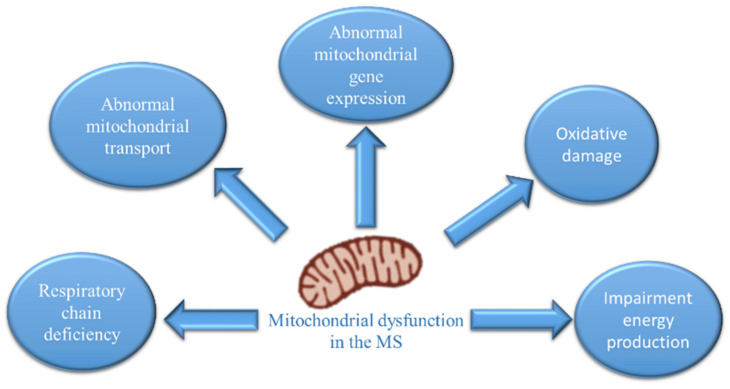
Changes in neuronal mitochondria in multiple sclerosis.

**Figure 2 ijms-23-02245-f002:**
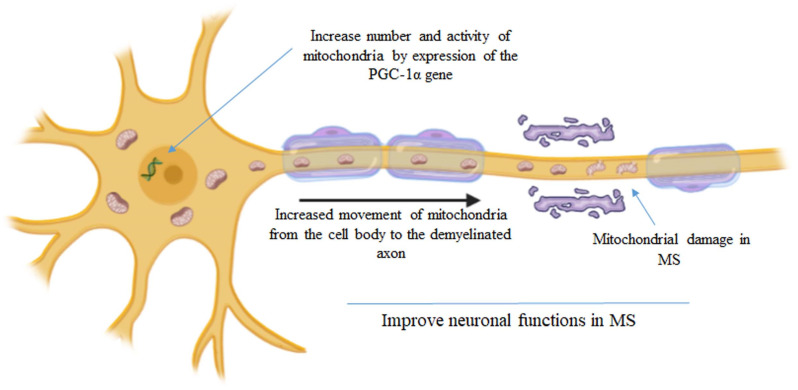
Increase in the mitochondrial number and activity represents a strategy to alleviate the energy imbalance in the demyelinated axon [10,11].

**Figure 3 ijms-23-02245-f003:**
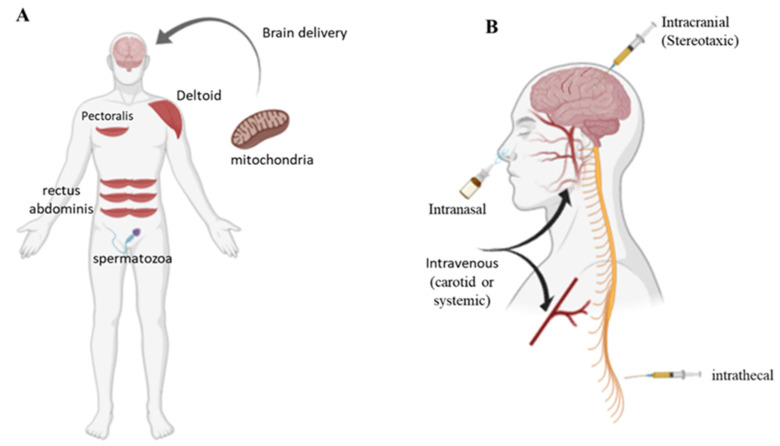
Sources (**A**) of human mitochondria and routes (**B**) for mitochondrial brain delivery.

**Figure 4 ijms-23-02245-f004:**
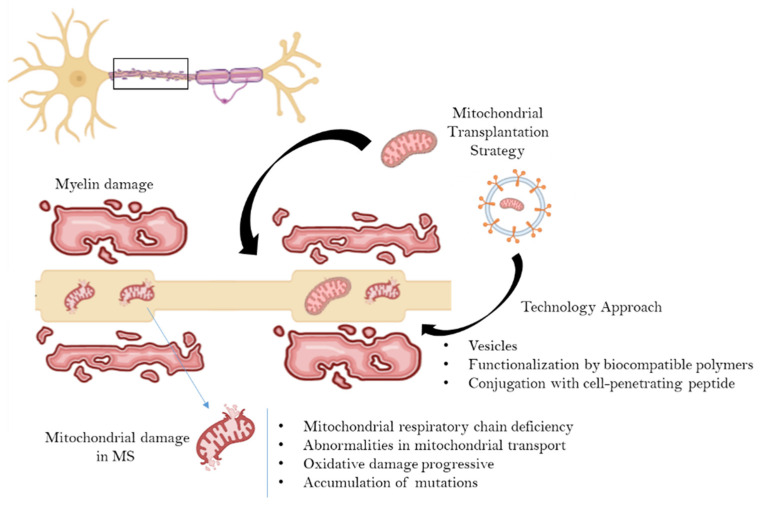
Schematic representation of the mitochondrial transplantation strategy in multiple sclerosis using a biotechnology approach based on encapsulation, protection and specific delivery of the healthy mitochondria in neuronal cells.

## Data Availability

Not applicable.

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
