# Peer review of "Promising Treatment for Multiple Sclerosis: Mitochondrial Transplantation"

_ijms, 2022, doi:10.3390/ijms23042245_

Round 1

Reviewer 1 Report

Your paper addresses a potential therapy focusing in MS management. This reviewer agrees the scientific landscape supporting this possibility should be further studied.

Author Response

We would like to thank the Referee for the appreciation of the manuscript.

Reviewer 2 Report

The work "Promising Treatment for Multiple Sclerosis: Mitochondrial Transplantation" presents an interesting approach on treatment of a neurodegenerative disease by supplying mitochondria on injury site.

In my opinion, the manuscript does not fit into the "Hypothesis" section. I acknowledge that it would have better value as an "Opinion" article.

For further information: https://www.mdpi.com/about/article_types.

The manuscript consists of a nice presentation of both mitochondria role in MS and mitochondria transplantation. Yet, it does not propose a novel view, an outstanding finding or interpretation. The authors have already published their experimental findings on the subject in "Picone P.; Porcelli G.; Bavisotto C.C.; Nuzzo D.; Galizzi G.; San Biagio P.L.; Bulone D.; Di Carlo M. Synaptosomes: new vesicles for neuronal mitochondrial transplantation. J. Nanobiotechnology 2021, 6, 19." Unfortunately, the present work is not as ambitious; it hardly delivers any extra insight.

I don't think it is suitable for publishing in a high impact scientific journal.

Author Response

In our article previously published (Picone et al., 2021) the use of synaptosomes as a delivery system for mitochondria was not proposed for specific neurological disease. In according with the referee, the sentence, which suggested that the cited publication referred to MS was deleted.

To confirm the novelty of our hypothesis, to date, no scientific article reported the possibility to use mitochondrial transplantation on MS. In fact, a network of knowledge research by PubMed "mitochondria transplantation and multiple sclerosis"  https://pubmed.ncbi.nlm.nih.gov/?term=%22mitochondria+transplantation+and+multiple+sclerosis%22 did not show results (quoted sentence not found: "multiple sclerosis and mitochondrial transplant").

We believe that this hypothesis, on the one hand, can attract the attention on mitochondrion as a potential target for MS (little considered), and on the other, suggest a potent innovative approach against MS (mitochondrial transplant) still not investigated by the community scientific.

Round 2

Reviewer 2 Report

These are excerpts from "Picone P.; Porcelli G.; Bavisotto C.C.; Nuzzo D.; Galizzi G.; San Biagio P.L.; Bulone D.; Di Carlo M. Synaptosomes: new vesicles for neuronal mitochondrial transplantation. J. Nanobiotechnology 2021, 6, 19.":

>>”synaptosomes ……….. used as a source of mitochondria when necessary for transplantation in mitochondria-damaged neuronal cells”

>>”synaptosome-mediated mitochondrial transplantation could be applicable for the treatment of many brain diseases in which traditional therapies have been unsuccessful”

>>”Taken together these results suggest that synaptosomes can be a natural vehicle for the delivery of molecules and organelles to neuronal cells. Further, the replacement of affected mitochondria with healthy ones could be a potential therapy for treating neuronal mitochondrial dysfunction-related diseases.”

>>”when damaged mitochondria cannot be replaced or restored, the possibility to transfer healthy mitochondria from one cell to another represents an attractive therapeutic strategy. Currently, the transfer of "alive" mitochondria into injured cells takes the name of mitochondrial transplantation and it is becoming a popular approach for the treatment of several diseases, including NDs “

The authors argue that in their previously published work they did not indicate a specific neurological disease. This is true. Yet, as seen in the above statements taken from their publication, they refer to “many brain diseases”, “neuronal mitochondrial dysfunction-related diseases”, “ND”.

In my opinion, they already proposed their scientific hypothesis. In this particular manuscript they chose MS out of all the possible “many brain diseases” or “ND”. This does not make their work original or novel. In the future, replacing MS with other “neuronal mitochondrial dysfunction-related diseases” would make for another scientific hypothesis?

Also, using quoted text when searching in Pubmed, most of the times, is going to return “quoted not found”. It depends on the exact particular text used as input for search. I recommend against using the quotation mark.

This is what I found on Pubmed:

https://www.ncbi.nlm.nih.gov/pmc/articles/PMC6465382/

https://pubmed.ncbi.nlm.nih.gov/33550783/

https://pubmed.ncbi.nlm.nih.gov/31210929/

In conclusion, I still don’t think this is an hypothesis article. The arguments the authors provided don’t stand up for scrutiny. I don’t think the manuscript is adequate for scientific publishing.